# Evaluation of Right Ventricular Function and Myocardial Microstructure in Fetal Hypoplastic Left Heart Syndrome

**DOI:** 10.3390/jcm11154456

**Published:** 2022-07-30

**Authors:** Jing Ma, Yaping Yuan, Li Zhang, Shizhen Chen, Haiyan Cao, Liu Hong, Juanjuan Liu, Xiaoyan Song, Jiawei Shi, Yi Zhang, Li Cui, Xin Zhou, Mingxing Xie

**Affiliations:** 1Department of Ultrasound, Union Hospital, Tongji Medical College, Huazhong University of Science and Technology, Wuhan 430022, China; d201981519@hust.edu.cn (J.M.); zli429@hust.edu.cn (L.Z.); caohaiyan2086@foxmail.com (H.C.); hongliu0826@hust.edu.cn (L.H.); d201981518@hust.edu.cn (J.L.); d201981524@hust.edu.cn (X.S.); shi_jw@hust.edu.cn (J.S.); zhangyievez@163.com (Y.Z.); cuil0805@163.com (L.C.); 2Clinical Research Center for Medical Imaging in Hubei Province, Wuhan 430022, China; 3Hubei Province Key Laboratory of Molecular Imaging, Wuhan 430022, China; 4State Key Laboratory of Magnetic Resonance and Atomic and Molecular Physics, National Center for Magnetic Resonance in Wuhan, Wuhan Institute of Physics and Mathematics, Innovation Academy for Precision Measurement Science and Technology, Chinese Academy of Sciences-Wuhan National Laboratory for Optoelectronics, Wuhan 430071, China; yuanyp@apm.ac.cn (Y.Y.); chenshizhen@wipm.ac.cn (S.C.); 5University of Chinese Academy of Sciences, Beijing 100049, China; 6Optics Valley Laboratory, Wuhan 430074, China

**Keywords:** cardiac diffusion tensor imaging, fetal echocardiography, hypoplastic left heart syndrome, myocardial microstructural, right ventricular function

## Abstract

Right ventricular (RV) function is one of the critical factors affecting the prognosis of fetuses with hypoplastic left heart syndrome (HLHS). Our study objectives included assessment of cardiac function and comprehensive measurement of cardiac microstructure. We retrospectively studied 42 fetuses diagnosed as HLHS by echocardiography. Myocardial deformation of the right ventricular wall was calculated automatically in offline software. Postmortem cardiac imaging for three control fetal hearts and four HLHS specimens was performed by a 9.4T DTI scanner. Myocardial deformation parameters of the RV (including strain, strain rate, and velocity) were significantly lower in HLHS fetuses (all *p* < 0.01). FA values increased (0.18 ± 0.01 vs. 0.21 ± 0.02; *p* < 0.01) in HLHS fetuses, but MD reduced (1.3 ± 0.15 vs. 0.88 ± 0.13; *p* < 0.001). The HLHS fetuses’ RV lateral base wall (−7.31 ± 51.91 vs. −6.85 ± 31.34; *p* = 0.25), middle wall (1.71 ± 50.92 vs. −9.38 ± 28.18; *p* < 0.001), and apical wall (−6.19 ± 46.61 vs. −11.16 ± 29.86, *p* < 0.001) had HA gradient ascent but HA gradient descent in the anteroseptal wall (*p* < 0.001) and inferoseptal wall (*p* < 0.001). RV basal lateral wall HA degrees were correlated with RVGLS (R^2^ = 0.97, *p* = 0.02). MD values were positively correlated with RVGLS (R^2^ = 0.93, *p* = 0.04). Our study found morphological and functional changes of the RV in HLHS fetuses, and cardiac function was related to the orientation patterns of myocardial fibers. It may provide insight into understanding the underlying mechanisms of impaired RV performance in HLHS.

## 1. Introduction

Hypoplastic left heart syndrome (HLHS) is a group of diseases mainly characterized by an underdeveloped left ventricle and no left ventricle apex, where the left-sided valves are usually dysplastic. The right ventricle (RV) dominates normal circulation, since the left ventricle cannot provide normal blood supply. RV dysfunction represents a strong determinant of poor prognosis in patients with HLHS [1,2,3]. It is essential to assess prenatal RV function in HLHS fetuses. Although, prenatally, the fetuses compensate by increasing output from the RV that should be stroked from the LV [4,5,6], the existing studies found abnormal fetal RV function using the SDI index and tissue Doppler technology [7,8]. Furthermore, normal right ventricular contraction is determined by the integrity of the free wall and interventricular septum [9]. Previous studies have shown that disorders of the arrangement of the myocardium and peripheral blood vessels, along with changes in the interstitial collagen components, occur in the left and right ventricles of HLHS patients [10,11,12,13]. However, all the specimens mentioned above were from postmortem neonates or children, so the right heart pathology cannot exclude the influence of hemodynamic changes or surgical intervention after birth. Although some studies have used fetal specimens, they have mainly focused on the pathological changes of the left-sided heart [14]. Therefore, there is still a lack of studies on the pathological changes of the right heart in prenatal HLHS patients, and traditional methods have limitations for comprehensive evaluation of pathological cardiac changes.

In recent years, cardiac diffusion tensor imaging has proven able to evaluate the orientation of myocardial fibers and the integrity of the myocardium using parameters such as helix angle (HA), transmural angle (TA), fractional anisotropy (FA), and mean diffusivity (MD). Many studies have shown that normal cardiac contraction and relaxation depend on the rearrangement of myocardial microstructures [15,16,17,18]. Ex-vivo imaging identified the normal cardiac sheetlets by the tertiary eigenvector of the diffusion tensor, supporting published results from histological studies [19]. Three-dimensional (3D) diffusion-tensor imaging (DTI) is a credible method to identify myocardial structural abnormalities.

We hypothesize that the myocardial architecture is altered in HLHS fetuses, which may be partly associated with reduced cardiac function. Hence, we assessed RV function in HLHS fetuses, and we used DTI to examine the cardiac microstructural differences in HLHS and normal specimens to explore the relationship between functional observations and myocardial microstructure dynamics.

## 2. Materials and Method

### 2.1. Study Population

We retrospectively studied 42 fetuses diagnosed with HLHS by echocardiography from 2016 to 2021. The inclusion criteria of HLHS were based on the guidelines for the management of neonates and infants with hypoplastic left heart syndrome [20]. Another 42 normal fetuses with matching gestational age were selected as a control group, defined as singleton fetuses with no structural cardiac anomaly. Exclusion criteria in both groups were as follows: (1) fetal complications with other serious heart malformations, such as double outlet right ventricle, endocardial cushion defects, and transposition of the great arteries; (2) twin or multiple pregnancy; (3) fetal growth restriction; (4) and pregnant women in the acute infection period. HLHS specimens were collected from pregnant women who were willing to donate fetuses in a fetal echocardiography study. Informed consent was obtained from all the families who donated fetal specimens. In the informed consent, families were informed that the specimens may be used for subsequent scientific research, and they all agreed. The ethics committee of Tongji Medical College of Huazhong University of Science and Technology and Union Hospital approved this study’s use of fetal specimens (S118,0510).

### 2.2. Fetal Echocardiography

A detailed fetal echocardiogram was performed by two experienced fetal echocardiography physicians using Voluson E10 and E8 ultrasound machines with eM6C and C2-9 transducers (2–9 MHz) (Tiefenbach 15, 4871 Zipf, Austria, GE Healthcare Austria GmbH & Co OG). Biventricular end-diastolic transverse diameters, length, foramen ovale (FO) diameter, ascending aorta (AAo) z-score, aortic valve (AV) z-score, main pulmonary artery (MPA) z-score, pulmonary valve (PV) z-score, left pulmonary artery (LPA) z-score, right pulmonary artery (RPA) z-score, and ductus arteriosus (DA) z-score were measured. The z-score values were all from Schneider et al [21]. The global sphericity index (GSI) was calculated by cardiac transverse diameters to length. The fetal heart blood flow, heart rate, and rhythm were observed and evaluated by M-mode echocardiography and color Doppler flow imaging. The velocity of the aorta, pulmonary artery, and ductus arteriosus were obtained by a spectral Doppler, and the peak systolic velocity of AV and PV flow, and the peak systolic and diastolic velocity of DA were measured at the three vessels and trachea (3VT) view [22].

### 2.3. Fetal Cardiac Function

We performed the right ventricular myocardial deformation from clips of four-chamber views in TOMTEC Imaging Systems (Unterschleissheim, Germany). Firstly, we manually set a single cardiac cycle according to the atrioventricular valve opening and closing activities; then, we stopped the frame at the end-diastolic period and manually placed three sampling points, respectively, in the endocardium of the right ventricular lateral wall and tricuspid annulus junction, the right ventricular plane of the interventricular septum and tricuspid annulus junction, and the middle of the right ventricular apex. The software automatically tracked the longitudinal movement of the right ventricular wall and obtained the longitudinal strain-time curve and strain-rate-time curve of the right ventricular wall. The right ventricular global longitudinal strain (RVGLS, %), right ventricular global longitudinal velocity (RVGLV, cm/s), right ventricular global longitudinal displacement (RVGLD, mm), and right ventricular global longitudinal strain rate (RVGLSR, *s*^−1^) were calculated automatically.

### 2.4. Post-Mortem CMR Data Acquisition

Before performing post-mortem imaging, fetal heart specimens were immersed in 4% formalin for 2 days; then, the hearts were stored in an oily solution (Fomblin Y-LVAC 6-06, Solvay Solexis, V.le Lombardia, Italy). CMR imaging was performed on a 9.4T scanner (Bruker BioSpin MRI GmbH, Ettlingen, Germany). Imaging parameters were as follows:

3D flash: Echo Time = 4.33 ms, Repetition Time = 30 ms, Averages = 16 ms, Pulse Angle = 25, Fov = 25 × 23 × 42 mm, Matrix= 256 × 256 × 360, Slice Thick = 42 mm, Scan Time= 10 h 18 m 29 s 760 ms; Diffusion tensor imaging: Echo Time = 16.5 ms, Repetition Time = 300 ms, Slice Thick = 42 mm, Fov = 25 × 23 × 42, FovCm= 2.5 × 2.3 × 4.2, SpatResol= 0.20 × 0.18 × 0.33 mm, Matrix = 128 × 128 × 128, Scan Time = 47 h 47 m 12 s 0 ms.

### 2.5. Post Imaging Process

Helix angle (HA), fractional anisotropy (FA), and mean diffusivity (MD) were defined as provided by the previous study [23]. Diffusion tensor cardiac imaging parameters were calculated on DSI studio software (developed by Fang-Cheng (Frank) Yeh, format date: accessed on 11 September 2020, https://dsi-studio.labsolver.org/). The right ventricle was divided into in the following 9 areas using ITK SNAP 3.8.0 software (developed by Paul A. Yushkevich and Guido Gerig, Philadelphia, PA 19104, USA) for regional HA analysis, which, namely, included 3 regions (lateral, anteroseptal, and infero-septal) on 3 cross-sections (basal, middle, and apical), according to the guidelines for the segment of cardiac regions [24]. Papillary muscle or chordae tendineae in the image were eliminated when the parameters were acquired.

### 2.6. Statistical Analysis

A statistical analysis was performed using SPSS version 23 (IBM; Armonk, New York, NY, USA) and GraphPad Prism 9.0.0 (121). Continuous variables were expressed as the mean ± standard deviation or the median (interval between quartiles). Categorical data were expressed as the number (%). The Shapiro–Wilk normality test was used for the normal distribution test, and the independent sample *t*-test was used for the comparison between the two groups of normal distribution. The Mann–Whitney U test was used to compare the non-normally distributed continuous variables between the two groups. The Kruskal–Wallis test was used for pairwise comparisons. The chi-square test or Fisher’s exact test was used for the inter-group comparison of classified variables. A univariate regression analysis and multivariate regression analysis were used to analyze the factors affecting fetal cardiac function in HLHS. Variables with *p* values < 0.05 in the univariate analysis were selected for the a multivariate regression. A Pearson correlation analysis was performed for an RV function and DTI parameters. Myocardial deformation parameters of 30 randomly selected fetuses for intra-observer and inter-observer reproducibility were calculated by a Bland–Altman analysis and intraclass correlation coefficients (ICCs) with 95% confidence intervals. A *p*-value < 0.05 was considered significant.

## 3. Results

Table 1 provides an overview of general clinical characteristics. Of the 42 fetuses with HLHS, a miniature LV was detected in 25 (59.52%) cases, silt-like LV in 15 (39.71%), and dilated LV in 2 (4.76%). Pathological phenotypes in HLHS were as follows: 11 (26.20%) with mitral atresia (MA)/aortic atresia (AA), 6 (14.29%) with mitral stenosis (MS)/AA, 10 (23.81%) with MS/aortic stenosis (AS), and 15 (35.71%) with a hypoplastic left heart complex (HLHC). The number of gravidities (3 (2–5) vs. 2 (1–3), *p* = 0.009) increased in the HLHS group. No difference was observed in other maternal characteristics between the two groups.

Table 2 demonstrates the cardiac morphological and hemodynamic differences between HLHS and the controls. There was a sharp drop in the LVEDD z-score (−3.51 (−5.58 to −2.1) vs. 0.42 (−0.13 to 0.83), *p* < 0.001), LVEDL z-score (−4.34 (−6.5 to −2.88) vs. 0.06 (−0.24 to 0.87, pp < 0.001), and RVEDL z-score (−0.83 (−1.64 to −0.23) vs. −0.03 (−0.58 to 0.24), *p* = 0.003) in HLHS fetuses, but the RVEDD z-score (0.66 (−0.34 to 1.75) vs. 0.04 (−0.27 to 0.51), *p* = 0.044) slightly increased in HLHS fetuses. While DA (0.62 (−0.29 to 1.04) vs. −0.12 (−0.41 to −0.12), *p* < 0.001) was broadening in HLHS fetuses, LPA and RPA were narrowed compared with the normal group, but there were no hemodynamic differences in HLHS fetuses.

The myocardial deformation parameters in HLHS fetuses, including RVGLS, RVGLVs, RVGLDd (mm), RVGLVd (cm/s), and RVGLSRs, were significantly reduced in HLHS fetuses (*p* < 0.01). More detailed data are shown in Table 3.

The univariate regression analysis for RV myocardial deformation parameters showed that the RVEDD/LVEDD (β, −0.48; [95% CI −1.81 to −0.25]; *p* = 0.01), LVEDD z-score (β, 0.37; [95% CI 0.06–0.92]; *p* = 0.03), and AAo z-score (β,0.39; [95%CI 0.05–0.69]; *p* = 0.02) were associated with a decreased RVGLS (Table 4). Variables with *p*-values < 0.05 in the univariate analysis were selected into a multivariate regression, so the left ventricular end diastolic diameter (LVEDD) z-score, ascending aorta (AAo) z-score, and ratio of right ventricular end diastolic diameter and left ventricular end diastolic diameter (RVEDD/LVEDD) were included in the multivariate regression analysis. However, only the LVEDD z-score (β, 0.48; 95%CI [0.08–1.01]; *p* = 0.02) was associated with reduced RVGLS, which indicated that the LVEDD z-score was the independent factor for reduced RVGLS in HLHS fetuses. (Table 5) (Figure 1).

Diffusion tensor imaging showed that FA values increased (0.18 ± 0.01 vs. 0.21 ± 0.02; *p* < 0.01) in HLHS fetuses (Figure 2), but MD reduced (1.3 ± 0.15 vs. 0.88 ± 0.13; *p* < 0.001) (Table 6) (Figure 3). We found that in HLHS fetuses, the RV lateral base wall (−7.31 ± 51.91 vs. −6.85 ± 31.34; *p* = 0.25), middle wall (1.71 ± 50.92 vs. −9.38 ± 28.18; *p* < 0.001), and apical wall (−6.19 ± 46.61 vs. −11.16 ± 29.86, *p* < 0.001) had an HA gradient ascent, which presented with the mean ± SD. However, the HA gradient descended in the anteroseptal wall (*p* < 0.001) and inferoseptal wall (*p* < 0.001) (Table 7) (Figure 4 and Figure 5).

Correlation analysis between RVGLS and HA degrees were performed. We found that only the RV basal lateral wall was correlated with decreased HA degrees [R^2^ 0.97(−0.99 to −0.44), *p* = 0.02] (Appendix A). Furthermore, MD values were positively correlated with RVGLS (R^2^ = 0.93, *p* = 0.04). No relationship between FA values and RVGLS was observed (Appendix A) (Figure 6).

The measurements of myocardial deformation all showed excellent reproducibility. Furthermore, the complete data of inter-observer variability and intra-observer variability are shown in Appendix A.

## 4. Discussion

In this study, the assessment of RV function and myocardial microstructure in fetal HLHS by two-dimensional echocardiography and diffusion tensor imaging were combined for the first time. We found that both RV systolic and diastolic dysfunction in HLHS and myocardial microstructures were different from those of normal fetuses.

Normally, according to the helical ventricular myocardial band (HVMB) theory, the right ventricle differs from the left ventricle in that it only contains circumferential fibers and a small number of ascending fibers. Contraction of the right ventricle is primarily determined by the longitudinal shortening of the sub-endocardium myocardium, radial movement of the RV lateral wall, and bulging of the interventricular septum into the RV [9,25]. In our study, an increased RVGSI, RVEDD z-score, and RVEDL z-score indicated that the right ventricle become more spherical. Furthermore, RVGLS (%), RVGLVs (cm/s), RVGLSRs (*s*^−1^), RVGLDd (mm), and RVGLVd (cm/s) of HLHS fetuses were all reduced compared with normal fetuses, suggesting abnormal RV diastolic and systolic function [26], similar to the findings of previous studies [5,7,8]. We found that the LVEDD z-Score was correlated with the decrease in RVGLS (%) by univariate and multivariate regression analyses. In the early stage, the influence of the left ventricle on right ventricular function mainly focused on left ventricular pathophysiology, and it was always believed that the pathological morphology of the left ventricle did not affect right ventricular function [27,28]. However, another study found that an increase in the end-diastolic left ventricular area after the Norwood procedure increased the risk of a heart transplant [29]. A study on prenatal HLHC found that higher LVEDL z-scores were an important predictor of Biventricular (BV) circulation [30]. Hence, we speculate that the dimension of the left ventricular end-diastolic stage is the main factor affecting RV function, rather than left ventricular pathological phenotypes. As previously mentioned, in HLHS fetuses, there was a decrease in the right ventricular longitudinal shortening ability, which was affected by the dimension of the left ventricle. Therefore, we hypothesized that there were changes in RV cardiac fibers. To prove this hypothesis, we evaluated the myocardial microstructure using CMR. Our results showed that the right ventricle FA value increased, and the MD value decreased in HLHS fetuses. FA values reflect the consistency of heart tissues, MD reflects the packing and integrity of the myocytes [16]. Previous studies found that myocardial MD values increased, and FA values decreased in patients with ischemic heart disease and hypertrophic cardiomyopathy, which may have been caused by a myocardial disorder and myocardial fibrosis [31,32,33]. However, the right ventricle of HLHS fetuses showed the opposite of the results of adult heart diseases, indicating that the right ventricle of HLHS fetuses became more compact and more organized. The RV lateral HA gradients ascend in HLHS fetuses. As previously mentioned, the normal right ventricle is not a global helix fiber, so HA gradient increases indicated lateral multiply oblique fibers in the RV. The normal interventricular septum is composed of oblique-oriented descending and ascending fibers [34,35]. However, decreasing HA gradients indicated that oblique fibers in the ventricular septum were eliminated. Our research results were consistent with those of a preliminary study, all illustrating a reduction in the circumferential muscles, which are right-ventricular-dominant in the lateral wall, but increased oblique fibers [36]. Hence, the structure of the right ventricle in HLHS fetuses may be closer to a global helix.

The correlation analysis indicated that RVGLS (%) was affected by MD values and basal lateral wall HA degrees. In the right ventricle basal lateral wall, increased HA gradients were beneficial to cardiac function in HLHS fetuses.

This study has certain limitations. Firstly, the investigation was retrospective in nature and used a single-center design. Furthermore, our data cannot be used for survival analysis, because most families chose to terminate pregnancy when fetuses were diagnosed as HLHS at the second trimester screening. It was really hard for us to collect the prognosis data. Deficiency of the sample size for the dilation phenotype resulted in the statistical analysis being unavailable. Secondly, the correlation analysis results were incomplete and limited by the heart specimens’ quantity, without further proving the direct relationship between structural tensor parameters and myocardial strain. We only described the causality between HA degrees in the RV basal lateral wall and RVGLS, and the interaction of HA degrees in remaining cardiac segments and RV function should be elucidated. Thirdly, we did not further validate the DTI results by histopathology. FA and MD values could be affected by myocardial cells or the myocardial matrix, so the mechanisms of DTI parameters’ variation are required for further study. Finally, our study performed the cardiac diffusion tensor imaging by 9.4T scanner, which currently was only available on specimens.

## 5. Conclusions

Our study found morphological and functional changes of the RV in HLHS fetuses, and cardiac function was related to the orientation patterns of myocardial fibers. It may provide insight into understanding the underlying mechanisms of impaired RV performance in HLHS.

## Figures and Tables

**Figure 1 jcm-11-04456-f001:**
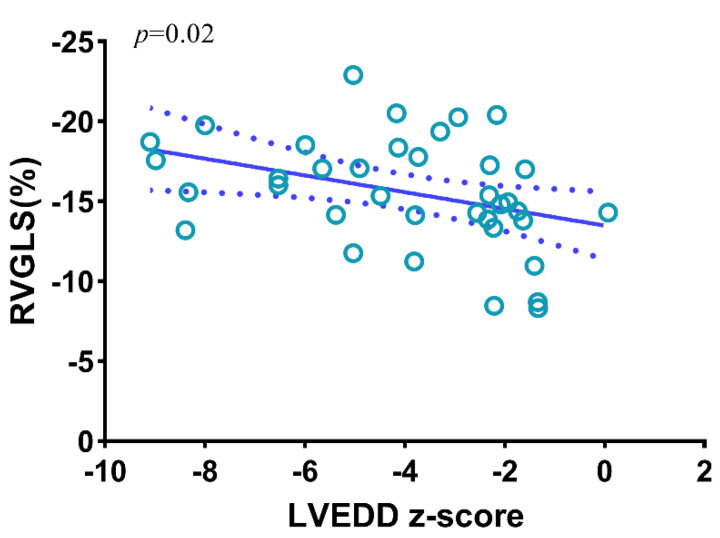
Correlation between RVGLS and LVEDD z-score. RVGLS, right ventricle global longitudinal strain; LVEDD z-score, left ventricular end diastolic dimension z-score.

**Figure 2 jcm-11-04456-f002:**
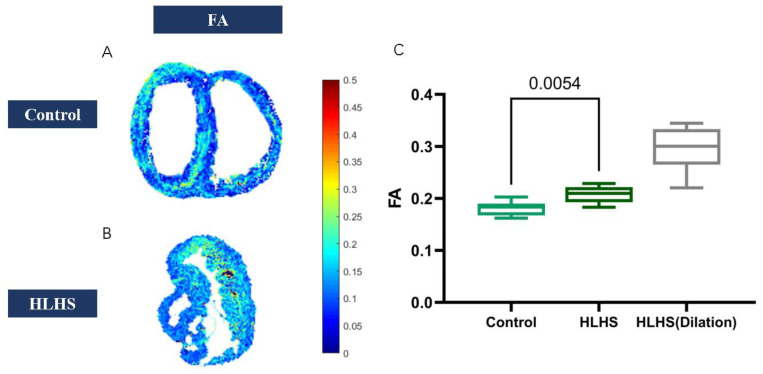
(**A**) FA map in normal fetuses. (**B**) FA map in HLHS fetuses. (**C**) FA values increased in HLHS fetuses.

**Figure 3 jcm-11-04456-f003:**
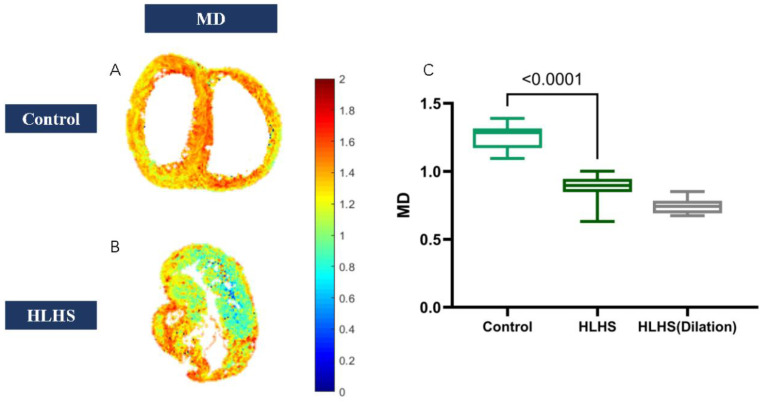
(**A**) MD map in normal fetuses. (**B**) MD map in HLHS fetuses. (**C**) MD values decreased in HLHS fetuses.

**Figure 4 jcm-11-04456-f004:**
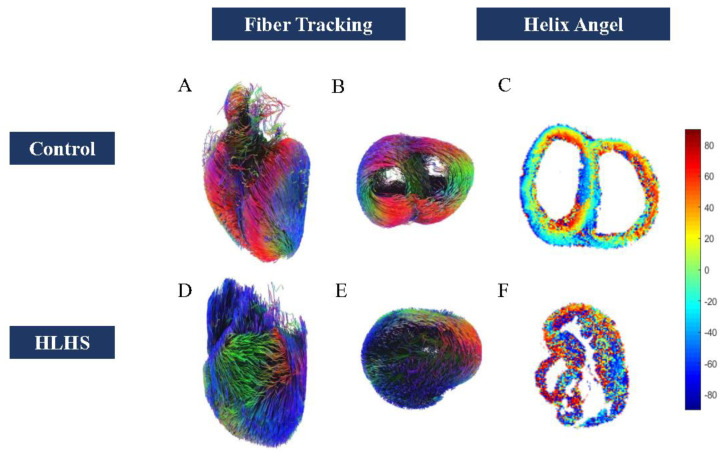
(**A**,**B**) Whole heart fiber tracking in a normal fetus. (**A**) from ventral view, (**B**) from apical view. (**D**,**E**) the whole heart fiber tracking in an HLHS fetus D from ventral view and (**E**) from apical view. (**C**) HA map in normal fetus. (**F**) HA map in HLHS fetus.

**Figure 5 jcm-11-04456-f005:**
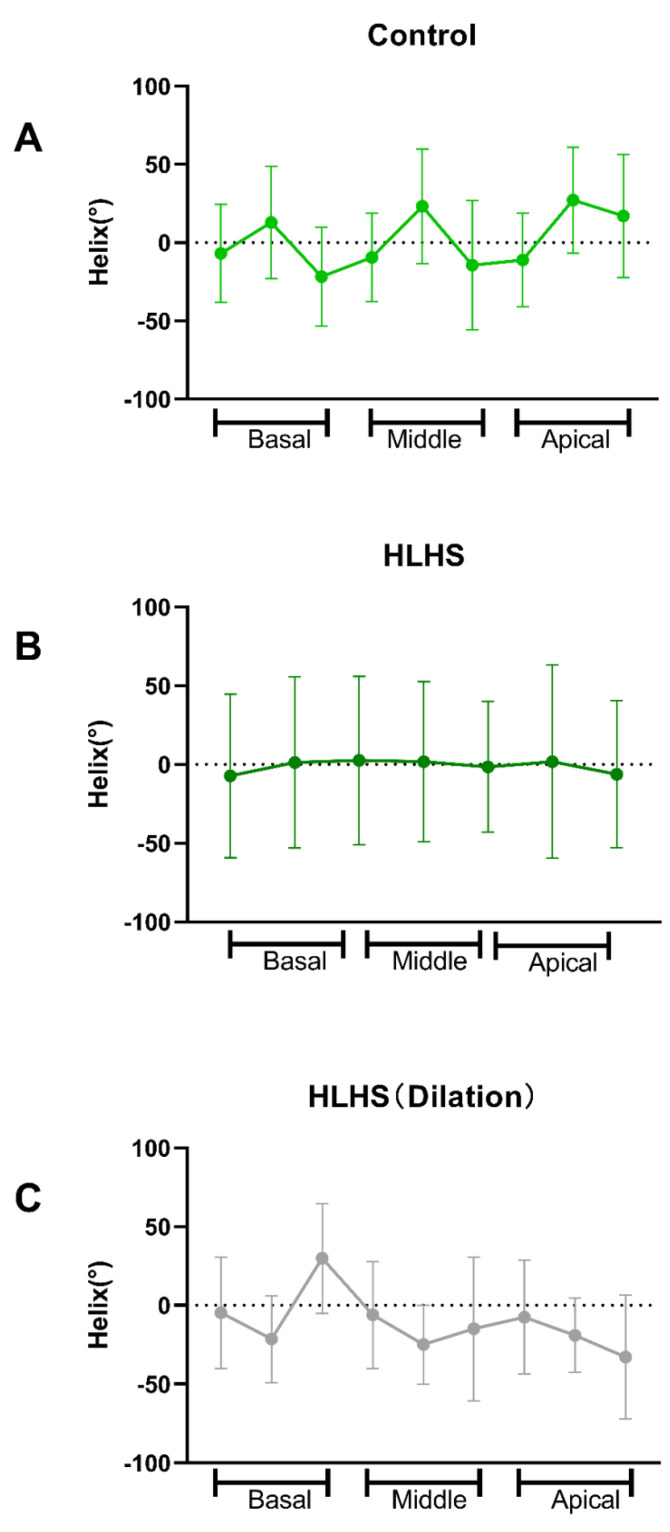
(**A**) HA gradient in normal fetus. (**B**) HA gradient in HLHS fetus. (**C**) HA gradient in an HLHS fetus with LV dilation.

**Figure 6 jcm-11-04456-f006:**
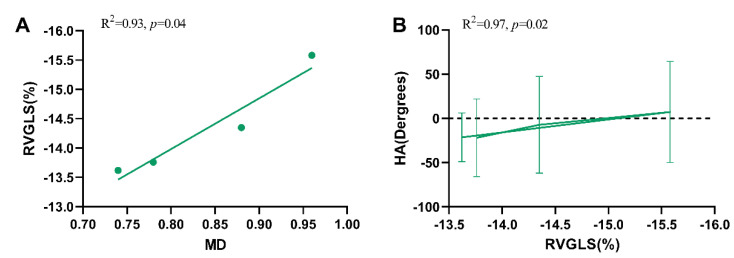
(**A**) Correlation between MD and RVGLS (%) in HLHS fetuses. (**B**) Correlation between HA degrees of the right ventricle basal lateral wall and RVGLS (%) in HLHS fetuses.

**Table 1 jcm-11-04456-t001:** Cohort characteristics.

Variable	Control Group (n = 42)	HLHS Group (n = 42)	*p* Vaule
Maternal characteristic			
Age (years)	30.00 ± 5.07	29.45 ± 5.46	0.40
BMI (kg/m^2^)	24.67 ± 3.34	23.40 ± 3.01	0.15
Gestational Age (weeks)	24.67 ± 3.34	24.49 ± 3.88	0.31
Obstetric history			
Gravidity	2(1–3)	3(2–5)	<0.01
Parity	1(0–2)	1(0–1)	0.05
Abortion	0(0–2)	1(0–2)	0.36
Anatomical variants			
Mitral atresia (MA)/aortic atresia (AA)		11(26.20%)	
Mitral stenosis (MS)/AA		6(14.29%)	
MS/aortic stenosis (AS)		10(23.81%)	
Hypoplastic left heart complex (HLHC)		15(35.71%)	
LV morphology			
silt-like		15(35.71%)	
miniature		25(59.52%)	
Dilation		2(4.76%)	

Continuous variables were expressed as the mean ± standard deviation or the median (interval between quartiles); categorical data were expressed as number (%). BMI, body Mass Index.

**Table 2 jcm-11-04456-t002:** Fetal left and right heart dimensions and hemodynamics in HLHS fetuses and normal fetuses at 23~30 gestational weeks.

Variable	Control Group	HLHS Group	*p*-Value
	(n = 42)	(n = 42)	
Morphometric measurements			
RVGSI	0.58 ± 0.08	0.71 ± 0.15	<0.01
LVGSI	0.49 ± 0.05	0.52 ± 0.15	0.02
LVEDD (cm)	0.95 (0.83 to 1.02)	0.56 (0.35 to 0.72)	<0.001
LVEDD z-score	0.42 (−0.13 to 0.83)	−3.51 (−5.58 to −2.1)	<0.001
LVEDL (cm)	1.92 (1.83 to 2.04)	1.04 (0.69 to 1.4)	<0.001
LVEDL z-score	0.06 (−0.24 to 0.87)	−4.34 (−6.5 to −2.88)	<0.001
RVEDD (cm)	0.97 (0.9 to 1.05)	1.10 (0.92–1.19)	0.02
RVEDD z-score	0.04 (−0.27 to 0.51)	0.66 (−0.34 to 1.75)	0.04
RVEDL (cm)	1.72 (1.57 to 1.83)	1.61 (1.34 to 1.74)	0.14
RVEDL z-score	−0.03 (−0.58 to 0.24)	−0.83 (−1.64 to −0.23)	<0.01
FO (cm)	0.48 ± 0.09	0.41 ± 0.15	0.03
PV z-score	0.19 (−0.47 to 0.97)	0.41 (−0.47 to 1.09)	0.69
Main PA z-score	0.65 (0.1 to 1.06)	0.89 (0.26 to 1.34)	0.12
LPA z-score	0.29 (−0.08 to 0.61)	−0.35 (−1.07 to 0.05)	<0.001
RPA z-score	0.12 (−0.39 to 0.57)	−0.56 (−1.21 to −0.06)	<0.001
DA z-score	−0.12 (−0.41 to −0.12)	0.62 (−0.29 to 1.04)	<0.001
AAo z-score	0.09 (−0.28 to 0.43)	−3.21 (−4.89 to −2.41)	<0.001
AV z-score	0.08 (−0.38 to 0.60)	−3.1 (−4.38 to −2.18)	<0.001
Hemodynamic Measurements			
Pulmonary artery			
Peak systolic velocity (cm/s)	70.3 ± 17.78	77.24 ± 14.96	0.14
Ductus arteriosus			
Peak systolic velocity (cm/s)	93.21 ± 22.99	96.51 ± 20.44	0.84
Ascending aorta			
Peak systolic velocity (cm/s)	94.43 ± 10.23	111.15 ± 49.45	0.04
FHR (bmp)	148.77 ± 7.07	147.37 ± 6.05	0.18

Values are expressed as mean ± SD or median (interquartile range); LVEDD z-score, left ventricular end diastolic diameter z-score; RVEDD z-score, right ventricular end diastolic diameter z-score; LVEDL z-score, left ventricular end diastolic length z-score; RVEDL z-score, right ventricular end diastolic length z-score; FO, foramen ovale; AAo, ascending aorta; AV, aortic valve; MPA, main pulmonary artery; PV, pulmonary valve; LPA, left pulmonary artery; RPA right pulmonary artery; DA, ductus arteriosus; GSI, global sphericity index; GA, gestational weeks. *p*-value means comparing the HLHS group with control group.

**Table 3 jcm-11-04456-t003:** Fetal right ventricle myocardial deformation by a two-dimensional speckle tracking echocardiography in HLHS fetuses and normal fetuses at 23~30 gestational weeks.

Variable	Control Group (n = 42)	HLHS Group (n = 42)	*p* Value
Systolic Function			
RVGLS (%)	−21.60 (−22.32 to −19.99)	−15.58 (−18.33 to −13.87)	<0.001
RVGLVs (cm/s)	−1.75 (−2.94 to −1.58)	−1.50 (−2.20 to −1.00)	0.02
RVGLSRs (*s*^−1^)	−1.95 (−2.45 to −1.65)	−1.40 (−1.68 to −1.20)	<0.01
Diastolic Function			
RVGLDd (mm)	1.90 (1.42 to 2.68)	1.20 (0.89 to 1.60)	<0.001
RVGLVd (cm/s)	1.60 (1.28 to 2.45)	0.95 (0.70 to 1.50)	<0.001
RVGLSRd (*s*^−1^)	2.05 (1.40 to 2.43)	1.80 (1.25 to 2.15)	0.10

Values are expressed as the mean ± SD or median (interquartile range); RVGLS, right ventricle global longitudinal strain; RVGLV, right ventricle global longitudinal velocity; RVGLD, right ventricle global longitudinal displacement; RVGLSR, right ventricle global longitudinal strain rate. *p*-value means comparing the HLHS group with the control group.

**Table 4 jcm-11-04456-t004:** Univariate regression analysis for RV myocardial deformation parameters in HLHS fetuses.

Variable	RVGLS		95% CI
	β	*p* Value	
GA (weeks)	0.03	0.85	0.19–0.85
RVEDD z-score	−0.26	0.12	−1.55–0.19
RVEDL z-score	0.15	0.38	−0.53–1.34
RVEDD/LVEDD	−0.48 *	0.01	−1.81 to −0.25
RVGSI	−0.24	0.14	−12.14 to 1.81
LVEDD z-score	0.37 *	0.03	0.06–0.92
LVEDL z-score	0.32	0.06	−0.02–0.75
LVGSI	0.079	0.64	−0.58–9.33
FO (cm)	−0.14	0.43	−9.69–4.18
LPA z-score	−0.21	0.21	−2.33–0.52
RPA z-score	−0.20	0.24	−2.47–0.65
DA z-score	−0.13	0.45	−1.51–0.68
AAo z-score	0.39 *	0.02	0.05–0.69
AV z-score	0.36	0.06	−0.02–0.89

LVEDD z-score, left ventricular end diastolic diameter z-score; RVEDD z-score, right ventricular end diastolic diameter z-score; LVEDL z-score, left ventricular end diastolic length z-score; RVEDL z-score; right ventricular end diastolic length z-score; AAo, ascending aorta; AV, aortic valve; MPA, main pulmonary artery; PV, pulmonary valve; LPA, left pulmonary arter; RPA right pulmonary artery; DA, ductus ateriosus; GSI, global sphericity index; FO, foramen ovale; GA, gestational weeks. CI, confidence interval. * *p* < 0.05.

**Table 5 jcm-11-04456-t005:** Multivariate regression analysis for RV myocardial deformation parameters in HLHS fetuses.

	RVGLS		
Variable	β	*p*	95% CI
RVEDD/LVEDD	0.06	0.90	
AAo z-score	0.20	0.34	
LVEDD z-score *	0.48	0.02	0.08–1.01

LVEDD z-score, left ventricular end diastolic diameter z-score; AAo, ascending aorta; RVEDD/LVEDD, ratio of right ventricular end diastolic diameter and left ventricular end diastolic diameter; CI, confidence interval. * *p* <0.05.

**Table 6 jcm-11-04456-t006:** Right ventricular myocardial microstructure dynamics in HLHS and control specimens at 24~30 gestational weeks.

Variable	Control (n = 3)	HLHS (n = 3)	HLHS (Dilation)	*p*-Value
FA	0.18 ± 0.01	0.21 ± 0.02	0.3 ± 0.04	<0.01
MD (×10^−3^, mm^2^·s^−1^)	1.3 ± 0.15	0.88 ± 0.13	0.74 ± 0.06	<0.001

Acquired by diffusion tension cardiovascular magnetic resonance technology. FA, fractional anisotropy, MD, mean diffusivity. *p*-values were compared with normal fetuses with HLHS fetuses.

**Table 7 jcm-11-04456-t007:** Regional characteristics of the helix angle gradient in the right ventricle in HLHS and control specimens at 24~30 gestational weeks.

Variable	Control (n = 3)	HLHS (n = 3)	HLHS (Dilation)	*p*-Value
Basal				
lateral	−6.85 ± 31.34	−7.31 ± 51.91	−4.77 ± 35.36	0.25
anteroseptal	12.89 ± 36.06	1.2 ± 54.31	−21.43 ± 27.59	<0.001
inferoseptal	−21.75 ± 31.63	2.54 ± 53.46	29.87 ± 34.93	<0.001
Middle				
lateral	−9.38 ± 28.18	1.71 ± 50.92	−6.07 ± 34.07	<0.001
anteroseptal	23.29 ± 36.59	−1.54 ± 41.38	−24.96 ± 25.05	<0.001
inferoseptal	−14.37 ± 41.38	1.73 ± 61.31	−14.94 ± 45.58	<0.001
Apical				
lateral	−11.16 ± 29.86	−6.19 ± 46.61	−7.53 ± 36.17	<0.001
anteroseptal	27.19 ± 33.72		−19.04 ± 23.58	
inferoseptal	17.06 ± 39.39		−32.85 ± 39.29	

Acquired by diffusion tension cardiovascular magnetic resonance technology. HA, helix angle; *p*-values were compared with normal fetuses with HLHS fetuses.

## Data Availability

The raw data supporting the conclusions of this article will be made available by the authors, without undue reservation.

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
