# Peer review of "Evaluation of Right Ventricular Function and Myocardial Microstructure in Fetal Hypoplastic Left Heart Syndrome"

_jcm, 2022, doi:10.3390/jcm11154456_

Round 1

Reviewer 1 Report

Review report-peer review v1-jcm 762639

Manuscript ID: jcm-1762639

Title: Objectives: Right ventricle (RV) function is the key to the prognosis

of fetuses with hypoplastic left heart syndrome (HLHS), our study objectives

included assessment of cardiac function by two-dimensional speckle tracking

and comprehensive ex vivo DT-CMR

Authors: Jing Ma, Shi zhen Chen, Ya ping Yuan, Hai yan Cao, Liu Hong, Juan

juan Liu, Xiao yan Song, Jia wei Shi, Yi Zhang, Li Cui, Xin Zhou *, Li Zhang

*, Mingxing Xie *

Submitted to section: Cardiology

Summary of manuscript: The aim was to study right ventricular function in fetuses diagnosed with hypoplastic left heart syndrome (HLHS) using two-dimensional speckle-tracking echocardiography (2DSTE) and DT-CMR post-mortem examination of the myocardial microstructure. The conclusion was both systolic and diastolic right ventricular dysfunction in HLHS fetuses, and the myocardial microstructure were altered compared to normal fetuses. Also, the shape of the right ventricle became more spherical in HLHS fetuses. The strength of the study was to combine the two methods 2DSTE and diffusion tension imaging (DTI).

Review 1 conclusion: This is an interesting and important descriptive retrospective study with the aim to evaluate fetuses with hypoplastic left heart syndrome and its prognosis of survival with ultrasound and magnetic resonance methodology.

The manuscript is not ready for publication, it needs major revision. It is hard to read and understand. The authors must help the readers to understand methodology, results with tables and figures.

These are my suggestions for improvement.

Title: Evaluation of Right Ventricle Function and Myocardial Microstructural in Fetal Hypoplastic Left Heart Syndrome.

Rev.v1: This sentence may have a typo, please change.

Abstract has too many words, write in one paragraph without headings, please rewrite

Keywords should be in alphabetic order, please change

Ethical statement from Institutional Review Board or Ethics committee is missing, please add

Informed Consent with statement from legal guardians is missing, please add

Method part is weak needs to be reworked and developed.

Line 43 ….left ventricle dysplasia…, the ventricle is underdeveloped and do not form the left ventricle apex, instead use left ventricle hypoplasia, the left sided valves are usually dysplastic, please rewrite the sentence.

Line 63 …cardiac contraction and diastole…., exchange diastole to relaxation?

Line 68  The manuscript would be strengthen of a clear hypothesis before the sentence that start with To understand….., please rewrite.

Line 90 Write out the abbreviation DA, please change

Line 94 The spectrum of the aorta, pulmonary artery, and ductus arteriosus were obtained by Spectral Doppler, it is better to use velocity in accordance with Doppler measurements, please rewrite.

Line 107-109 please add the units for Strain, Velocity, Displacement and Strain Rate.

Results

Line148 The text does not conform to table 1, please rewrite

Line 153 Table 2 demonstrate the differences between HLHS and the controls, please rewrite

Table 1.  The headings of the table must be explanatory, please add. See above line 148 and in footnote, number (%), please add. Suggestion heading, cohort characteristics

Table 2. The headings of the table must be explanatory, please add. Suggestion heading, Fetal left and right heart dimensions and hemodynamics in HLHS fetuses and normal fetuses at gestational week (range).

Table 3. The table shows, Fetal right ventricle myocardial deformation by two-dimensional speckle tracking echocardiography in HLHS fetuses and normal fetuses in gestational week …(range). There is no unit for RVGLSR. Please add.

Table 4-5. Tables assume HLHS fetuses in correlation to normal fetuses, please add in heading if so, table 5 footnote is not correct, describe the factors included in the multifactor analysis.

Table 6. Heading must be clarified, suggestion Fetal right ventricular myocardial microstructure dynamics in HLHS fetuses and normal fetuses in gestional week…(range?) in footnote acquired by diffusion tension cardiovascular magnetic resonance technology.

Table 7. Heading must be clarified, suggestion Fetal right ventricular myocardial microstructure function (or characteristics) in HLHS fetuses and normal fetuses in gestational week….(range?)  in footnote acquired by diffusion tenson cardiovascular magnetic resonance technology.

Tables 8-10 These tables could be added to supplemental material

Figures 1-6 Correct font in figure 1. Keep figures and reduce number of tables.

Discussion

Line 264. There is abbreviation to write out.

Reviewer 2 Report

It not clear if this is a retrospective or prospective study. Did you have consent for post mortem CMR? 

It is not clear to me the role of post mortem CMR in this paper, in a very subgroups of patients (3 controls and 4 HLCS). I suggest to consider to analyze only US.

Conclusion: please rewrite the sentence. There is no meaning.

Reviewer 3 Report

This manuscript presents a very sophisticated sonographic method to prove that  fetal heart with HLHS is different from normal heart anatomy and function. I would suggest to compare HLHS who did survived with HLHS group who did not survived 1 stage. Than these information would be usefull for readers

Round 2

Reviewer 1 Report

Review report-peer review v2-jcm 762639

Manuscript ID: jcm-1762639

Title: Evaluation of right ventricular function and myocardial microstructure in fetal hypoplastic left heart syndrome

Objectives: Right ventricle (RV) function is the key to the prognosis

of fetuses with hypoplastic left heart syndrome (HLHS), our study objectives

included assessment of cardiac function by two-dimensional speckle tracking

and comprehensive ex vivo DT-CMR

Authors: Jing Ma, Shi zhen Chen, Ya ping Yuan, Hai yan Cao, Liu Hong, Juan

juan Liu, Xiao yan Song, Jia wei Shi, Yi Zhang, Li Cui, Xin Zhou *, Li Zhang

*, Mingxing Xie *

Submitted to section: Cardiology

Summary of manuscript: The aim was to study right ventricular function in fetuses diagnosed with hypoplastic left heart syndrome (HLHS) using two-dimensional speckle-tracking echocardiography (2DSTE) and DT-CMR post-mortem examination of the myocardial microstructure. The conclusion was both systolic and diastolic right ventricular dysfunction in HLHS fetuses, and the myocardial microstructure were altered compared to normal fetuses. Also, the shape of the right ventricle became more spherical in HLHS fetuses. The strength of the study was to combine the two methods 2DSTE and diffusion tension imaging (DTI).

Review 2 conclusion: This is an interesting and important descriptive retrospective study with the aim to evaluate fetuses with hypoplastic left heart syndrome and its prognosis of survival with ultrasound and magnetic resonance methodology.

The authors have improved the manuscript to help the readers to understand methodology, results with tables and figures.

These are my suggestions.

Abstract has to many words. See authors instruction. I suggest using your previous abstract to shorten to 200 words. Only the most important information should be described and concluded with the most important findings.

Line 34 Abstract, in brackets remove etc. You have to many decimals in the results.

Line 65 Keywords, misspelling

Line 93, I think you mean, …. normal cardiac contraction and relaxation depend on….

Line 173 Correct font

Line 191 Statistical analysis, classification variables were expressed by absolute person-time. Have you calculated absolute person time? if not, delete this sentence.

Line 211 Increasing gravidity? do you mean number of pregnancies/gravidities?

Table 1. All figures should be declared with figures. In footnote units is missing for anatomical variants numbers (%).

Check font and font size.

Kindly regards

Reviewer 2 Report

The revised manuscript reads better and the authors have addressed precisely all my comments. 

Author Response

Thank you for your nice comments. We improved our manuscript accroding to your previously valuable suggestions.